# Jak Inhibitors for Treatment of Autoimmune Diseases: Lessons from Systemic Sclerosis and Systemic Lupus Erythematosus

**DOI:** 10.3390/ph15080936

**Published:** 2022-07-28

**Authors:** Przemysław Kotyla, Olga Gumkowska-Sroka, Bartosz Wnuk, Kacper Kotyla

**Affiliations:** 1Department of Internal Medicine, Rheumatology and Clinical Immunology Medical University of Silesia, 40-635 Katowice, Poland; kacper.kotyla@gmail.com; 2Department of Rheumatology, Voivodeship Hospital No. 5, 41-200 Sosnowiec, Poland; oag@poczta.onet.pl; 3Department of Rehabilitation, Faculty of Health Sciences in Katowice, Medical University of Silesia, 40-635 Katowice, Poland; bwnuk@sum.edu.pl

**Keywords:** systemic lupus erythematosus, systemic sclerosis JAK inhibitors, cytokine network

## Abstract

Systemic sclerosis and systemic lupus erythematosus represent two distinct autoimmune diseases belonging to the group of connective tissue disorders. Despite the great progress in the basic science, this progress has not been translated to the development of novel therapeutic approaches that can radically change the face of these diseases. The discovery of JAK kinases, which are tyrosine kinases coupled with cytokine receptors, may open a new chapter in the treatment of so far untreatable diseases. Small synthetic compounds that can block Janus kinases and interact directly with cytokine signalling may provide therapeutic potential in these diseases. In this review, we discuss the therapeutic potential of Jak kinases in light of the cytokine network that JAK kinases are able to interact with. We also provide the theoretical background for the rationale of blocking cytokines with specific JAK inhibitors.

## 1. Introduction

Connective tissue diseases are a group of chronic diseases with an autoimmune background. Recent advances in genetics, pathology, and clinical immunology have started to explain the potential mechanisms responsible for the initiation and propagation of these diseases [1,2]. Unfortunately, with a few exceptions, this progress has not translated to the development of new disease-specific drugs that can interact with these key immunological disease-critical mechanisms. In fact, the result of treatment is still based on non-specific immunosuppression realized mainly via steroid and cytotoxic drug administration [3,4]. The result of this immunosuppression is the reduction of the central and peripheral activity of the dysregulated immune system. Among the many not fully elucidated mechanisms leading to the restoration of the proper function of the immune system, cytokine activity is believed to play an important role [5]. Cytokines are soluble intercellular crosstalk transmitters, which are responsible for modulating immune system functioning. However, in the setting of immune dysregulation, cytokines become the executive arm of autoimmunity directly responsible for maintaining the autoimmune response. This is especially true for inflammatory arthropathies, where the role of some proinflammatory cytokines is well established. At the end of the last century, the understanding of the role of some proinflammatory cytokines, e.g., TNFα, IL-1, IL-6, or IL-17, translated to the development of high-affinity molecular antibodies blocking these cytokines’ function and halting disease progression.

At that time, almost all the scientific papers on rheumatoid arthritis started with the sentence “TNF is a key cytokine in RA development”, suggesting that we had finally found the ‘holy grail’ and that we would be able to successfully treat all inflammatory conditions [6,7,8]. With the progression of research, it became clear that blocking only one cytokine is not enough to stop an autoimmune response and that the plethora of cytokines, chemokines, and intercellular signals cannot be stopped with only one drug. Moreover, despite several similarities in the clinical pictures, rheumatic conditions differ between each other in terms of their pathophysiological background and mechanisms of inflammatory response. Therefore, one effective drug in a given rheumatic disease does not work in all others and vice versa. That was the strong impulse for the identification of disease specific mediators and the invention of the drugs capable of inhibiting them. Indeed, the progress in understanding the pathophysiological background of some rheumatic diseases translated to the development of anti-cytokine drugs that proved to be efficacious in the treatment of many (but not all) aspects of inflammation. These drugs, commonly referred to as biological disease modifying drugs (bDMARDs) or more commonly as biologics, revolutionized the treatment of inflammatory arthritides. The mode of action of biologics is based mainly on blocking the inflammatory cytokines; however, other mechanisms have been successfully used, such as depleting the population of antibody producing B-cells and interfering in the co-stimulation of immunocompetent cells. Unfortunately, blocking one cytokine with specific biologics is sometimes clinically infective; in addition, the treatments can lose their efficacy over time due to immunogenicity or the activation of the other signalling pathways, thereby bypassing the cytokine already blocked.

Despite the therapeutic efficacy of biological DMARDs, it has become evident that treatment with bDMARDs has several limitations; thus, not all patients may benefit from such a treatment. Moreover, biologics are large proteins that are difficult to synthetize, and the parenteral route of administration is often an obstacle for patients. Treatment with biologics can produce adverse drug reactions such as tuberculosis, heart failure, neuropathies, and others [9,10,11].

In the early 1990s, the discovery of a family of intracellular tyrosine kinases attached to several cytokine receptors resulted in the further discovery of the pathway that transmits signals from a cytokine to the nucleus. The fact that the role of this discovery was not completely understood explains the term initially used to characterize them—“just another kinase”.

The discovery of the immune pathway that orchestrates the immune mechanism translated to the quest for new therapeutic approaches. Among several mechanisms that transmit cytokine signals to the nucleus, the JAK/STAT pathway is of special interest, as it is responsible for transmitting signals from many cytokines within the same signaling pathway.

Recently, a new class of low-weight compounds capable of blocking several cytokines was developed and tested in chronic conditions including rheumatoid arthritis, psoriatic arthritis, and haematological diseases. These drugs are commonly referred to as JAK kinase inhibitors or (Jakinibs). To understand the role of these specific cytokines we must be aware that these low molecular weight proteins or glycoproteins may orchestrate not only the peripheral immune system but also act in the central phase of the immune response when antigen or autoantigen recognition takes place. During the recognition of an antigen presented by antigen presenting cells to a naïve T cell, a plethora of cytokines acting as co-stimulatory signals is released. Moreover, the activated immunocompetent cells can synthesize and release other cytokines that regulate the survival, development, and function of other immune and non-immune cells. Cytokines signal via a wide variety of receptor structures categorized into several receptor superfamilies. After their interaction with the extracellular domain of the receptor, they can activate long chains of transmission molecules to activate specific genes in the nucleus.

## 2. Cytokine Signalling Pathways

The essential role in transmitting cytokine signals is played by protein kinases attached to the intracellular part of the receptors. Cytokine signalling and the regulation of their activity is realized via the interaction between cytokines, chemokines, and growth factors commonly referred to as ligands and the extracellular domain of the given receptor. Several types of receptors are involved in this process and are usually categorized into receptor subfamilies. Among them we may distinguish the TNF receptor subfamily, the IL-1 subfamily receptors, and the IL-17 and Janus kinase-associated receptors. The TNF receptor subfamily signal via TNFR and transmit their signal by further utilizing transmission molecules such as TRADD, TRAF2, and RIP1, resulting in the activation of NF-κB and MAPK signalling and the subsequent gene activation and expression of pro-inflammatory cytokines, such as interleukin 6 and 8 (IL-6 and IL-8) [12,13]. IL-1 subfamily receptors transmit signals from the IL-1 family (IL-1, IL-33, and IL-36) to the nucleus through the use of MyD88 and IRAK adaptor proteins [14,15]. The other important proinflammatory signals are transmitted via IL-17. Upon interaction with its receptors, IL-17 activates multiple signalling cascades resulting in the activation of the NF-κB, C/EBPβ, C/EBPδ, and MAPK pathways [16]. However, considering the number of cytokine signals that are transmitted, the Janus kinases associated with cytokine specific receptors play an important role by interacting with more than 50 cytokines belonging to the class I and class II superfamily cytokines [17].

As the JAK kinases can be easily blocked with small synthetic compounds, JAK kinases are a promising target to halt cytokine signalling and restore immune balance. This hypothesis was successfully tested, and several JAK inhibitors were introduced to common clinical practices demonstrating their safety and efficacy in the treatment of several hematologic conditions and inflammatory arthropathies (rheumatoid arthritis, psoriatic arthritis, and spondyloarthropathies) [18]. Taking into account the many similarities between inflammatory arthropathies and connective tissue diseases, it would be reasonable to establish whether there is a role for JAK kinase inhibitors in the treatment of connective tissue diseases (CTDs) [19,20]. This finding may be especially important considering the lack of accepted therapies for treating connective tissue diseases. Since inflammation, orchestrated by a network of pro- and anti-inflammatory cytokines, plays an unequivocal role in the development of several CTDs, new therapeutic strategies targeting the inflammatory and signalling pathways may offer promising opportunities.

## 3. Structure and Function of JAK/STAT Pathway

JAKs belong to the family of tyrosine kinases. Currently, four JAKs have been identified: JAK1, JAK2, JAK 3, and TYK2. The structure of JAK is composed of seven homologous regions (JH1-JH7) forming four structural domains (FERM, SH2, Pseudokinase, and Kinase domains). The JH1 and JH2 regions are located at the C terminal end of an enzyme-encoding kinase and a pseudokinase, respectively. In contrast to JH1, the JH2 homology region is characterized by dual kinase activity and regulates catalytic kinase activity, simply limiting the ligand-independent catalytic activity of the kinase domain.

The remaining four regions do not have catalytic activity and serve as a harbour to the cytoplasmatic tails of receptors. They build two domains: FERM, where F stands for F4.1 protein, E for ezrin, R for radixin, and M for moesin (JH5-JH7), and the SH2 domain (Src homology-2), such as (JH3-JH4). The activation (phosphorylation) process starts with the binding of the cytokine to its receptors followed by the dimerization of the receptor subunits that place receptor-associated kinases in close proximity to each other and thereby facilitate their mutual activation. When activated, JAK kinases further transmit signals to the intracellular space by activating the transcription factors known as STATs. These translocate to the nucleus to modulate the promoter region of specific genes and orchestrate transcription [21]. The four JAKs work together to form homo or heterodimers that partially explain their signalling specificity. When activated, JAKs transmit their signal and activate STAT proteins. At the current level of knowledge, seven STATs have been identified: STAT1, STAT2, STAT3, STAT4, STAT5a, STAT5b1, and STAT6 [22]. STAT consists of an N terminal domain, a coiled tail domain, an SRC-homology 2 domain, a DNA-binding domain, a linker domain, a phosphotyrosyl tail, and a transactivation domain located at the C terminus [23,24]. Each of the STAT’s domains play a unique role. The N terminal, a conserved domain, is responsible for the STAT’s phosphorylation. The DNA-binding domain, usually located between amino acids residue 400- and 500-, forms a complex of DNA and STAT proteins, while the SH2 domain’s function is to interact with other proteins. Finally, the C-terminal domain plays a role as an activation centre for the whole STAT molecule [25]. The role of the JAK/STAT system is crucial for the proper functioning of the immune system. Several cytokines and growth factors signal in this way. JAK-associated receptors are classically categorized as class I and class II receptor families. The typical structure of the receptor consists of one to four receptor chains. They form the extracellular cytokine R homology domain (CHD) and cytokine-binding domain. The difference between class I and class II receptors is the presence of two disulphide bridges linking the cysteines in the two chains of class I receptors [26]. However, the most important difference is the expression of a highly conservated Trp- Ser-Xaa-Trp-Ser WSXWS motif in the class I molecule, which is absent in the receptors of type II cytokines [27,28]. Class I receptors transmit signals from four cytokines families and hormone-like ligands. IL-2, IL-4, IL-7, IL-9, IL-15, and IL-21 transmit their signals via gamma chain receptors (γc) [29]. The beta family receptor is responsible for transmitting signals from GM-CSF, IL-3, and IL-5 [30]. The third class of receptor is built with the gp 130 protein (or its homologue) and transmits signals from IL- 6, IL-11, IL-31, IL-35, and IL-27 [31]. The last member of the Class I receptor subfamily interacts with IL-12 and IL-23, which are heterodimeric cytokines that share a common p40 subunit [32]. Class I receptors are also used by hormone-like cytokines such as erythropoietin, growth hormones, leptin, or thrombopoietin [33]. Contrary to this, class II receptors are responsible for transmitting signals from interferons and the IL-10 cytokine family (IL-10, IL-19, IL-20, IL-22, IL-24, and IL-26) [34].

The classical pathway by which cytokines transmit their signals is based on the JAK/STAT pathway. However, signals from activated JAK may utilize some by-pass pathways. In detail, JAK kinase is a direct activator of the PI3K/AKT signalling pathway and JAK, when phosphorylated, activates PI3K [25].

The activity of the JAK/STAT pathway is negatively regulated by several mechanisms aimed at limiting cytokine signalling and thus reducing the cytokine response. This process is mediated via the activation of specific regulatory sentences in the nucleus to express regulatory factors such as the suppressor of cytokine signalling (SOCS), the protein of activated STAT (PIAS), and protein tyrosine phosphatase (PTP) [35]. The SOCS family consists of eight members: cytokine-inducible SH2-containing protein (CIS) and SOCS1, SOCS2, SOCS3, SOCS4, SOCS5, SOCS6, and SOCS7. They contribute to the regulation of the immune response. CIS and SOCS1-3 negatively regulate cytokine signalling via the JAK/STAT pathway, while SOCS4-7 inhibits growth factor-mediated signalling [17]. The primary target of SOCS1-3 activity is a JAK molecule. SOCS molecules can interact with the JAK catalytic centre (SOCS1), directly inhibit receptor subunit (SCOCS3), or compete with the STAT molecule to form a receptor complex [36].

Unlike SOCS, PIAS proteins are expressed constitutively. Their role is to regulate the intensity of apoptosis, cell survival, and tissue renewal. The main mode of action of the four known PIAS proteins (PIAS1-4) is to control gene expression, which appears to be performed through the controlling activity of several transcriptional regulators [37].

## 4. Connective Tissue Diseases—The Role of Cytokine Network

Systemic sclerosis

Systemic sclerosis (SSc) is a connective tissue disease characterized by massive skin fibrosis vasculopathy and internal organs’ involvement leading to terminal organ dysfunction. Although skin and internal organ fibrosis is a hallmark of SSc, these changes are secondary to an aberrant innate and adaptive immune system activation and uncontrolled cytokine release [38,39]. The pathophysiological and immune-mediated mechanism leading to the onset and progression of the disease are not fully elucidated. Apart from the role of the known and established fibrosis-driving factors such as TGF-β, PDGF, ET-1, and IGF1 and the chemokines MCP-1/CCL2, MIP-1α/CCL3, MIP-1β/CCL4, and IL-8/CXCL8 [40], some studies suggest the role of the Th2 immune response and the subsequent release of Th2-dependent cytokines such as IL-4, IL-5, and IL-13, which are known to control fibrotic processes [41]. Additionally, similar to systemic lupus erythematosus and other connective tissue diseases, patients with SSc showed an overexpression of IFNα, suggesting a direct pathogenetic role in the disease’s development [42]. Importantly, the IFN signature can be detected at very early stages of the disease (many years before a formal diagnosis can be established), suggesting that IFN upregulation is an early event and may contribute significantly to the disease pathogenesis [43].

The next main players in the field of SSc pathogenesis are the IL-6 and IL-6 cytokine family. IL-6 has an established role in the pathogenesis of SSc as it is responsible for vasculopathy and driving the fibrotic processes. It correlates with disease activity and the extent of skin thickening [44,45,46].Thus, this correlation was the pathophysiological background for the clinical trials aimed to block IL-6 activity. Although promising, the results of the studies completed thus far have not achieved their primary endpoints [47,48,49].

The role of the IL-6 cytokine family was recently substantiated in several studies on other IL-6 family members. In line with this, de Almaiida et al. reported elevated serum levels of soluble oncostatin M receptor (sOSMR) and sgp130 in patients with systemic sclerosis that correlated with the presence of digital ulcers and negatively correlated with oesophagus dysfunction [50]. Moreover, as recently shown by Marden et al., OSM signalling may play an important role during vessel degeneration and fibrosis in patients with SSc [51]. The next member of the IL-6 family, IL-31, is synthesized by activated Th2 cells and is widely expressed by many other cells including macrophages, keratinocytes, and fibroblasts [52]. As a strong regulator of Th2 function, Il-31 may perpetuate the fibrotic process [53] and be responsible for the predominance of the Th2 response observed in SSc patients [54]. Scanty data exist regarding the role of another strong proinflammatory cytokine, namely, IL-12. Although the role of this cytokine seems to be of lesser importance in systemic sclerosis, a member of the IL-12 subfamily, IL-35, identified almost 25 years ago, has recently attracted high interest. So far, the role of IL-35 in the pathogenesis of systemic sclerosis is a matter of academic dispute. Recently, it was shown that IL-35 was elevated in SSc patients, where it mainly acts as an anti-inflammatory cytokine reducing CD4 T cell differentiation and facilitating Treg induction [55]. In contrast, other studies suggest the proinflammatory activity of IL-35 and its role as a potent profibrotic factor [56]. Even more conflicting data exist regarding the role of two other members of the IL-12 subfamily, namely, IL-23 and IL-27. In laboratory studies, IL-23 has been proven to be a potent inducer of collagen type I in dermal fibroblasts, while IL-27 showed only a moderate effect [57].

The role of the typical Th2-dependent cytokines IL-4 and IL-13 in SSc may be partially explained through the context of a strong profibrotic effect that these cytokines drive. Interleukin-4 and IL-13 are overexpressed in the skin and serum of SSc patients, and they directly stimulate collagen synthesis in fibroblasts and drive Th cell polarization toward a Th2 response with strong profibrotic effects [53,58,59]. Less is known about the role of anti-inflammatory cytokines belonging to the IL-10 superfamily. Utilizing the Scl-cGVHD model, an animal model for human SSc, it was shown that IL-10–producing Bregs were able to suppress skin fibrosis [60]. Moreover, in patients with SSc, IL-10–producing Bregs have been found to be reduced and correlated with disease activity, but not with SSc-specific antibodies [61,62,63]. There are scanty data on the role of the other IL-10 family cytokines in the development of systemic sclerosis. The studies completed to date have shown the reduced expression of IL-20 or dysregulated IL-23 signalling as potential mechanisms for uncontrolled collagen deposition in skin and internal organs, suggesting the antifibrotic potential of these anti-inflammatory cytokines [64,65]; however, the precise role of IL-20 and IL-23 is poorly understood Figure 1.

Systemic sclerosis is characterized by the activation of IL-12 cytokine family that exert a mainly profibrotic effect. Different types of receptors signalling via the JAK/STAT pathway may be potentially modulated by the inhibition of receptor-attached JAK. IL-4, IL-5, and IL-13 cytokines belonging to the γ chain receptor subfamily exert a profibrotic effect that may be blocked via JAK-1 and JAK-3 inhibitors, resulting in the reduced polarisation of Th cells toward a profibrotic Th2 response. Similarly, the IL-12 cytokines, IL-23 and IL-27, are characterized by a significant profibrotic effect. The role of the last member of the Il-12 family, IL-35, is characterized by dual pro-fibrotic and anti-fibrotic activity. Therefore, the net effect depends on which signalling pathway is predominantly blocked. JAKis can block signalling via the IFN receptor that translates to a reduction in the IFN signature (and a potential therapeutic effect). IL-10 cytokine family members exert both anti-fibrotic (IL-10) as well as strong profibrotic effects (IL-31). In line with this, the inhibition of the IL-10 family’s signalling may exaggerate the pro-fibrotic effect when the IL-10 signalling is blocked. IL-6 family cytokines, especially IL-6 and IL-31, are recognized as strong profibrotic agents. Blocking JAK coupled with the IL-6 type receptor may result in a direct therapeutic effect.

## 5. Do JAKi Offer Therapeutic Potential in SSc?

Despite the enormous progress in genetic, clinical, and experimental immunology, systemic sclerosis is still a condition where no disease specific treatments exist. As a result, its treatment is directed toward the protection of vitally important internal organs’ function using untargeted immunosuppression.

With the emerging role of cytokines and interferons driving the inflammation and fibrotic processes in SSc patients, it would be reasonable to test whether JAK inhibitors show any therapeutic potential for this disease. This may be especially worthwhile for IFNs, as the upregulation of IFN-α is central to the pathogenesis of the disease. SSc patients are expected to benefit from therapies that neutralize IFN-α, reduce its production, or block its downstream effects [66]. This might be achieved by blocking JAK kinases (JAK1 and Tyk2) attached to IFNR. The hypothesis on the usefulness of JAK inhibitors is currently undergoing testing in three clinical trials from China (Baricitinib- NCT05300932), France (Ruxolitinib-NCT04206644), and the USA (Tofacitinib- NCT03274076). The completed study from the USA did not show the superiority of Tofacitinib versus a placebo towards skin improvement (measured as a change in mRSS), nor an improvement in CRISS (Combined Response Index Systemic sclerosis). This is in contrast to previously published data, where in small observational studies Tofacitinib contributed to the reduction of skin thickness in SSc patients measured both clinically [67] as well as with ultrasound [68]. Obviously, it is too early to draw final conclusions. Considering that non-selective Jakinibs may block both proinflammatory and profibrotic cytokines as well as exert a negative impact on those showing anti-inflammatory and antifibrotic potential, it would be reasonable to test the other inhibitors’ activity towards this condition.

## 6. Systemic Lupus Erythematosus

Systemic lupus erythematosus (SLE) is a connective tissue disease that serves as a prototype for other autoimmune diseases. SLE is characterized by the activation of the immune system by multiple self-nuclear antigens, leading to antibody formation and subsequent antigen–antibody complex formation (immune complexes—ICs) [69]. This is the first step in the inflammatory response of the immune system [70]. These processes are augmented by ineffective apoptosis [71] and the defective clearance of apoptotic cells, leading to the release of self-antigens that may be easily recognized by the host immune system [72]. With a defect in the function of the innate immunity, characterized by reduced phagocytosis [73], there is an accumulation of plasmacytoid dendritic cells (pDCs) in the inflamed tissues, as well as the improper functioning of the complement system [74,75]. SLE is an autoimmune disease where all the components of the immune system may be affected. Moreover, parallel to the defective functioning of the innate immune system, there is dysfunction in adaptive immunity characterized by the increased activity of B cells, defects in the removal of auto-reacting B cells [76], and the hyperactivated phenotype of T cells can increase the generation of autoAbs [77,78]. This process is orchestrated by a plethora of cytokines released by immunocompetent cells [79,80]. Therefore, the targeting of specific cytokines may be a promising option to restore the proper function of the immune system. Considering the therapeutic success of BAFF and interferon targeting as the approved therapeutic modalities in SLE, it is reasonable to halt the signalling of other cytokines in the hope that it would restore the functioning of the immune system [79,81]. This approach can be reached by the use of JAK inhibitors as they target many proinflammatory cytokines including interferons [82].

With multiple organs’ involvement and potential damage to all the vitally important organs (brain, kidneys, lungs, skin, the hematopoietic system, joints, vessels, etc.), the clinical picture of the disease is complicated. This may be explained by the complexity of several immune mechanisms involved in the disease. Glucocorticoids and conventional immunosuppressants are the cornerstone of therapy, but their targets are non-specific, and the severe side effects can limit their usage in a substantial portion of patients; therefore, more effective, safe, and targeted therapies are needed.

## 7. Interferons in the Pathogenesis of SLE

The interferon Type I is now recognized as one of the key cytokines orchestrating the autoimmune processes in SLE, and they bridge the innate and acquired autoimmune response commonly observed in the course of the disease. Since the first observation of elevated levels of IFN in SLE in 1979, hundreds of subsequent studies have confirmed the role of this cytokine and linked it with the expression of thousands of IFN-related genes [83]. The presence of this phenomena is commonly referred to as an interferon signature; however, this is not restricted to SLE, as it may be observed in other autoimmune diseases [84,85,86]. Recently, Haynes et al. described a set of 93 genes whose expression seems to be linked to SLE, thus helping to distinguish SLE from the other INF-related diseases, other autoimmune diseases, neoplasms, and infections [87]. IFNs are synthesized in response to the activation of plasmacytoid dendritic cells (pDC CD11c-CD123^high^). The pDC represent only 1% of all cells but they seem to play a crucial role in SLE development as they express TLR7 and TLR9 in their exosomes [88]. In the context of SLE, TLR7 and TLR9 play an essential role as they can detect cell-derived single stranded RNA as well as unmethylated CpG dsDNA [89,90]. pDCs are characterized by the constitutive expression of the transcription factor IRF7, which enables them to synthesize a large amount of IFN type I in response to RNA and DNA nucleic acids [91].

Although the term interferon signature is commonly used in the context of IFN type I-related gene expression, the other interferons [92], namely type II and III, play a role in the pathogenesis of SLE [93,94]. Moreover, these specific types of interferons are linked to the various forms of lupus presentation, making the disease’s presentation complex and its treatment challenging [95].

In detail, type I interferon activity is linked to haematological disease presentation (anaemia, leukopenia, and thrombocytopaenia), mucocutaneous presentation, and the development of lupus nephropathy [96,97]. In contrast, type III interferons are responsible for the formation of antiphospholipid antibodies [94]. Less is known about the role of IFN type II in this regard. The activity of IFN II is not related to any specific disease presentation. However, IFN gamma is a unique interferon, as it is mainly synthesized by Th1 and NK cells, and its role should be recognized in the context of the activation of the immune system, which sometimes precedes the development of clinically overt disease as recently demonstrated by Liu et al. [98]. The authors identified 143 differentially expressed genes (DEGs) in patients with SLE naive for treatment. Most of the identified genes were upregulated and responsible for the activation of the immune system [98].

Therefore, the role of specific interferons should be discussed in the light of their mutual interactions rather than the activity of one specific cytokine.

## 8. Interleukin-6 in Lupus

IL-6 has been found to be elevated in SLE patients and correlated with disease activity [99]. This might be a theoretical argument for IL-6 inhibition as a therapeutic approach in SLE patients, using biologics to target IL-6 or its receptor or the administration of JAK inhibitors to target JAK molecules attached to the IL-6 receptor. The levels of IL-6 in sera, joint fluid, urine, and cerebrospinal fluid in patients with SLE are high [100,101,102]. However, at the moment, the role of IL-6 inhibition in SLE is a matter of controversy, as no clinically important therapeutic effects have been observed with IL-6 inhibition [103]. So, it is possible that the therapeutic effect of JAK inhibitors is not mediated by the inhibition of IL-6 signalling.

## 9. IL-2: The Role in Lupus Development

Inerleukin-2 (IL-2), a pleiotropic cytokine belonging to the wide family of γ-chain cytokines, and it is released mostly by conventional T cells upon stimulation [104]. It is a crucial factor for T cells’ survival and development as well as for the polarisation of T cells toward Treg [105]. The role of IL-2 should be recognized in terms of autoimmunity caused by IL-2 deficiency [106]. Therefore, halting IL-2 signalling with JAK inhibitors may aggravate the disease course. However, this may not be true as the action of JAK inhibitors, as it may be indirect and depend on inhibition of other cytokines negatively influencing IL-2 levels. That may be especially true for IL-23, a cytokine that signals via the JAK/STAT system, which has been shown to suppress IL-2 levels [107]. On the other hand, IL-2 signalling could promote IFNγ production [108] and the enhanced expression of IL-12 receptors [109]. So, at the moment, the role of IL-2 as a target for JAK inhibitors is still controversial and more studies are required to clarify the role of IL-2 signalling and its activity in lupus management.

## 10. IL-12 and IL-23 in Lupus

Two cytokines, IL-12 and IL-23, belonging to the IL-12 family, recently attracted high attention as possible causative factors in the development of SLE. This is largely due to the fact that both cytokines represent a group of strong proinflammatory cytokines and their role has been already established in several autoimmune and inflammatory disorders [110].

Although both cytokines have similar structures and share a common receptor subunit p40 (together with p35 and p19 for IL-12 and IL-23, respectively), the role of these cytokines in the differentiation of naïve T cells is different. While IL-12 exerts a strong effect on naïve T cells to promote differentiation toward Th1, IL-23 is responsible for Th polarisation toward a Th17 response [111]. Both cytokines utilize the JAK/STAT pathway to signal with the subsequent activation of STAT1, STAT3, STAT4, and STAT5; however, the activation of the heterodimer JAK2/TYK 2 predominantly translates to the phosphorylation of STAT 4 for IL-12 and STAT3 for IL-23 [112,113,114,115].

The Th1 response driven by IL-12 translates to the activation of natural killer cells (NK), cytotoxic pathways, and the production of IFN by dendritic cells [116,117].

Patients with SLE are characterised by high levels of IL-12 and IL-12-related cytokines with the component p40, and this has been found to correlate positively with disease activity and negatively with serum complement concentration [118,119].

The importance of IL-12 signalling was recently established in a clinical trial with ustekinumab, a monoclonal antibody targeting the p40 subunit shared by IL-12 and IL-23 [120]. In the study, the patients randomized to ustekinumab showed an improvement in disease activity as measured with the SLEDAI scale, an improvement in skin status, and a reduction of swollen and tender joints count. This study confirmed the importance of IL-12/IL-23 axis in the development of SLE; however, due to the relatively small group sizes it should be interpreted cautiously.

## 11. IL-10 and IL-10 Cytokine Family in SLE

IL-10 is typically an anti-inflammatory cytokine; thus, it is surprising that the other cytokines belonging to this family sharing a similar cytokine structure and receptor exert quite different properties and are recognized as a factor driving inflammation. Structurally, the IL-10 family is further divided into three subfamilies [121]. The first one encompasses IL-10 itself, the second group (IL-20 subfamily) consists of IL-19, IL-20, IL-22, IL-24, and IL-26, while the third contains type III interferons.

IL-10 exerts a potent anti-inflammatory effect, targeting monocytes and macrophages and blocking the release of inflammatory cytokines [122]. Acting directly on antigen-presenting cells, it reduces the expression of MHC class II molecules as well as costimulatory molecules. Furthermore, it blocks Th cells’ polarisation toward Th1 response by inhibiting IL-12 and IL-23 signalling [123]. So, considering these properties, application of JAK inhibitors, attenuating IL-10 signalling may shift the balance toward a proinflammatory response.

Data from clinical studies and animal models suggested that IL-20 may play a pathogenic role in the development of lupus nephritis [124]. In one study, the expression of IL-20 and its receptors have been shown to be upregulated in SLE mice compared to control animals [125]. Renal IL-20 overexpression was also observed in lupus patients. These observations suggest the direct role of IL-20 in the development of SLE.

Even more pronounced pathogenic effects were observed with regard to IL-22, which was found to be elevated in the sera of SLE patients and correlated with disease activity [126], although not all studies confirmed this result [127].

The next member of the IL- 20 family, IL-26, is mainly synthesized by Th1 and Th17 memory cells. The role of IL-26 is largely unknown; however, scanty data suggest that elevated levels of Il-26 observed in SLE patients may contribute to the activity of the disease. Recently, IL-26 has been proposed as a potential marker of SLE activity [128] Figure 2.

Type I interferons play a crucial role in the development of SLE; thus, the inhibition of JAK attached to an IFN receptor may explain the therapeutic effects of JAKis. IL-2, a cytokine belonging to the γ chain receptor family, is responsible for Treg development; therefore, it exerts an anti-inflammatory effect that may be blocked by IL-23. Typical proinflammatory cytokines of the IL-12 family are responsible for the polarisation of Th cells toward Th1 and Th17. As the direct peripheral role of these cytokines is unknown, the inhibition of this pathway is probably indirect (blocking inflammatory Th1 and Th17 response). The IL-10 family receptor transmits both anti-inflammatory (IL10) and pro-inflammatory signals (IL-20, IL-22 and IL-26). Therefore, JAKi administration may exert a therapeutic effect when predominantly proinflammatory cytokines are inhibited.

## 12. Jakinibs for Systemic Lupus

A plethora of cytokines are involved in the pathogenesis of SLE, acting directly on effector cells or creating a proinflammatory milieu. This impedes our understanding of the role of one specific cytokine, and it is not possible to predict the clinical effect when one specific cytokine is blocked. Moreover, with respect to JAK inhibitors, the application of the drug may potentially block several pathways that may not necessarily contribute to the restoration of immune imbalance. In addition, the fact is that one especially non-specific Jakinib can block several forms of cytokine signalling. These limitations should be kept in mind when JAK inhibitors are used for the treatment of SLE. On the other hand, one inhibitor could block several proinflammatory pathways and simply switch off the signalling from multiple cytokines. That was the hypothesis when testing the role of JAKi in real world clinical practice. Firstly, considering the many similarities between RA and SLE, it is plausible that the inhibition of JAK may at least halt SLE-related synovitis. Indeed, a non-specific JAK inhibitor, Tofacitinib, has been shown to halt the signalling of the JAK/STAT pathway, resulting in the reduction of IL-17 and IFNγ and the proliferation of CD4+ T cells, with the subsequent suppression of IL-6 production by RASFs and IL-8 synthesis by CD14+ cells and decreased structural cartilage damage [129]. Direct testing of the JAKi role in SLE started with the study with MRL/*lpr* lupus-prone mice. In this study, treatment with Tofacitinib reduced disease activity (nephritis, mucocutaneous presentation, and autoantibody synthesis). Moreover, treatment with Tofacitinib contributed to the reduction of proinflammatory cytokines and interferon expression. Tofacitinib could also restore endothelium damage and dysfunction [130]. Parallel to this, several case reports and small observational studies indicated the potential of Tofacitinib to reduce disease activity [131,132,133,134,135,136,137,138]. These promising results were recently substantiated in patients with SLE treated with Tofacitinib. As it was shown in an ex vivo model with CD4 T cells from patients with SLE, pre-treatment with tofacitinib resulted in the restoration (inhibition) of distorted Th cells’ function via enhancing the expression of TGFβRI. It is plausible that the inhibition of IL-6-signalling realized by the inhibition of a Jak kinase attached to an IL-6 receptor may play a role in this process.

A recently published, randomized, double blind, and placebo-controlled trial of Tofacitinib (5 mg twice a day) in patients with SLE JAK inhibitors showed a satisfactorily safety profile, improved lipid profile disturbances, decreased IFN type I signature, and restored endothelial function. However, the Authors failed to show any statistically significant changes in reduction of diseases activity, as it was clearly stated that the study was not aimed to test the drug’s efficacy [139].

Administration of a more specific Jakinib, Baricitinib, may bring even more therapeutic opportunities. A selective inhibitor for JAK1 and JAK2, approved for use in rheumatoid arthritis, recently demonstrated its potential as an agent for the treatment of lupus patients. In an animal MRL/Mp-*Fas^lpr^* (MRL/*lpr*) mice model of lupus, Baricitinib significantly suppressed lupus-like phenotypes of MRL/*lpr* mice, such as splenomegaly, lymphadenopathy, proteinuria, and immune system activation including autoantibodies formation and pro-inflammatory cytokines’ release. It also regulated immunocompetent cells’ activity and effectively reduced renal inflammation. In this in vitro phase of the study, Baricitinib negatively influenced B cell differentiation and restored the disrupted cytoskeletal structures of podocytes under inflammatory stimulation by blocking the JAK/STAT pathway [140]. Those promising data were verified in a double-blind placebo-controlled study of 314 lupus patients randomly assigned to receive baricitinib at 2 mg per day, 4 mg per day, or a placebo. At the end of the study at week 24, 70% patients receiving Baricitinib 4 mg achieved resolution of SLEDAI 2 k arthritis or rash [141]. JAK usage is potentially linked to a reduction of all (or almost all) cytokines’ signalling via the JAK/STAT pathway. To test this, Dörner et al. conducted a trial aimed to check the expression of key cytokines related to lupus. At week 12, Baricitinib 4 mg significantly reduced levels of C-C motif chemokine ligand (CCL) 19, C-X-C motif chemokine ligand (CXCL) 10, tumour necrosis factor alpha (TNF-α), TNF receptor superfamily member (TNFRSF)9/CD137, PD-L1, IL-6, and IL-12β [142]. Additionally, the authors observed a suppression of cytokines related to IFN I activities that translated to the reduction in the concentration of dsDNA antibodies, an improvement in SLEDAI 2000 scale, and a reduction in swollen and tender joints [142]. Some discrepancy occurs regarding JAKi selectivity. As various JAK/TYK2 combinations may serve as signal transducers from different types of receptors, the net cytokine effect may vary between different types of JAKi used. In general, when specific TYK2 inhibitors are used, they predominantly block signalling from IL-12 but also inhibit signalling from IFNα. On the other hand, the application of a specific JAK2 inhibitor blocks signalling mainly from IFNγ, in contrast to panJAKi (Tofacitinib) that blocks activation mediated by IFNα and IFNγ [143]. Disappointing data come from a recently completed trial with Filgotinib in patients with cutaneous lupus erythematosus. The study did not meet the primary endpoint as the patients treated with Filgotinib did not significantly improve their CLASI score [144]. However, the results from this study are not surprising. Filgotinib is a JAK1-specific inhibitor targeting almost all SLE-cytokine related receptors, but it is not able to stop the signalling of IL-12/IL-23, IL3, and IL-5. Blocking IL12/IL-23 signalling might be considered essential in cutaneous lupus given the results with ustekinumab in the treatment of SLE. Therefore, a lack of IL-12/IL-23 inhibition may translate directly to the failure of the study. As far as the role of IL-3 is concerned, we may only speculate that IL-3 may be involved in development of SLE indirectly acting together with IFNs to create a dual IL-3/IFN gene signature [145].

Ruxolitinib, a JAK1 and JAK2 kinase inhibitor approved by the FDA for the treatment of myelofibrosis, showed a potential to attenuate severe skin changes in a mouse model of SLE [146]. Following this promising result, the team from Rochester University started to recruit patients for a 12-week study with Ruxolitinib cream applied topically to areas with an active lupus skin lesion. The results from this study will be available soon. There are several ongoing trials registered at clinical trial registers in the USA and Europe (clinicaltral.gov and European Trial database) shown in Table 1. The results from these trials may provide a precise insight into how JAK inhibitors work in the settings of systemic sclerosis and lupus. At the current time, we may only hypothesise that some JAK inhibitors may show a satisfactory safety/efficacy profile, potentially enabling them to be registered as treatments for lupus.

## Figures and Tables

**Figure 1 pharmaceuticals-15-00936-f001:**
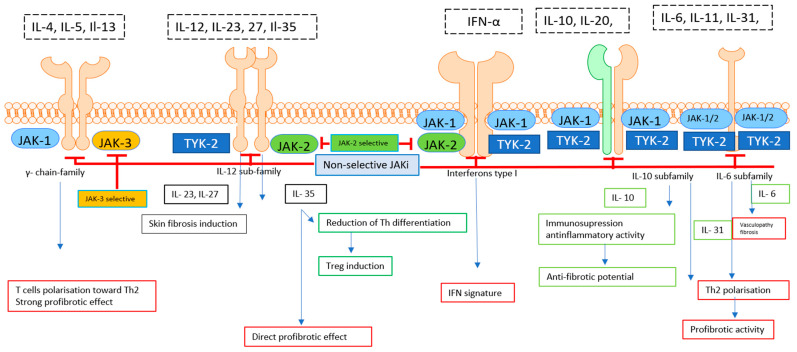
Cytokine network in systemic sclerosis.

**Figure 2 pharmaceuticals-15-00936-f002:**
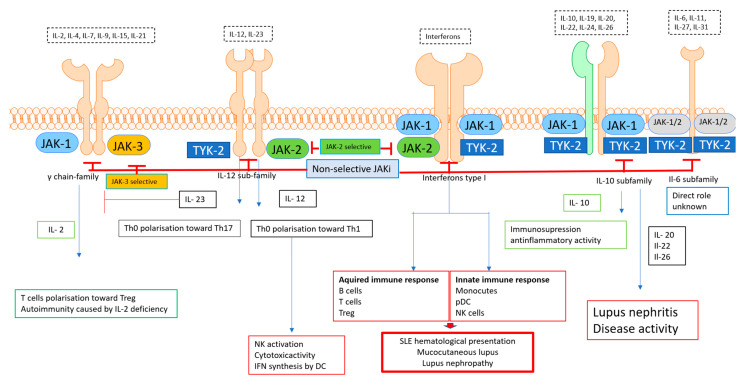
The role of cytokines in SLE.

**Table 1 pharmaceuticals-15-00936-t001:** List of Jakinibs with therapeutic potential in treatment of Systemic Lupus and Systemic Sclerosis.

JAK Inhibitor Name	Selectivity	Indication (NCT Study)
Tofacitinib	Non-selective	Tested for lupus treatment (NCT02535689, NCT05048238, and NCT03288324)
Baricitinib	Non-selective	Studies in Lupus terminated (NCT03616912, NCT03843125)
Ruxolitinib	Non-selective	Trial in DLE (NCT04908280)
Peficitinib	Non-selective	Tested for RA treatment
Filgotinib	Jak-1 selective	Assessed for treatment of CLE (NCT03134222)
Upadacitinib	Jak-1 selective	Evaluated for lupus treatment (NCT04451772 and NCT03978520)
Solcitinib	Jak-1 selective	Study in SLE terminated NCT01777256
Itacitinib	Jak-1 selective	Under investigation in Systemic Sclerosis (NCT04789850)
AC430	Jak-2	Potential role in the treatment of cancer and autoimmune diseases
TG101209	JAK-2	Potential role in the treatment of leukaemias and myeloproliferative disorders
Decernotinib	JAK-3	Tested for treatment in RA
R 333	Jak-3	Further studies terminated
PF 06651600Ritlecitinib	JAK-3 (dual JAK-3/TEC inhibitor)	Evaluated in alopecia areata, RA
Brepocitinib	JAK-1/Tyk2	Tested in SLE (NCT03845517)
Deucravacitinib	Tyk-2	Assessed in SLE (NCT03252587) (NCT03920267)

TEC tyrosine kinase expressed in hepatocellular carcinoma; DLE discoid lupus erythematosus; CLE cutaneous lupus erythematosus.

## Data Availability

Data sharing not applicable.

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
