# Peer review of "Jak Inhibitors for Treatment of Autoimmune Diseases: Lessons from Systemic Sclerosis and Systemic Lupus Erythematosus"

_pharmaceuticals, 2022, doi:10.3390/ph15080936_

Round 1

Reviewer 1 Report

The manuscript is very interesting and well structured. I recommend its publication in the present form. 

I just suggest performing extensive editing of the English language and style.

Author Response

Thank you very much for your valuable comments We do believe  that they contribute significantly to improvement of quality of our manuscript

As You proposed we added table that summarises recent trials  in SLE and SSc.

Many thanks for finding typo errors they were corrected as  follow:

  Row 30  “…is to reduce of… changed to The final result of this immunosuppression is reduction of

Row 50 in treatment in many…”(typo) changed to  efficacious in treatment of many

  Row 55 “….biologics is sometimes infective….”(please clarify) we changed to is sometimes clinically infective

 Row 58   check punctuation        punctuation corrected

  Row 108 “…. propagation ….”(please clarify) word propagation removed The current sentence sounds like cytokines plays an unequivocal role in the development of several CTDs

Row 160 check punctuation punctuation corrected

Row 205 “…Less in known….” (typo) we wrote less is known

   Rows 231-232 check punctuation and repetition of words we rewrote to systemic sclerosis is still a condition where no disease-specific treatment exists.

  Row 307 ” ….corelated…”(typo) changed to correlated

 Row 316 “….vide…..”(typo) corrected to wide

   Row 342 “…corelate …”(typo) corrected to correlate

Row 405 “….a key role seems to be…(please check the verb and the sentence) We wrote It is plausible that inhibition of IL-6 signalling that is realized by inhibition of Jak kinase attached to an IL-6 receptor may play the role in this process.

Row 410 check punctuation punctuation corrected

 Rows 438-439  check the sentence (“but not these…” )and block letters for IL-12 and IL-23 capaiatl letters used additionally we wrote Filgotinib is a JAK1 specific inhibitor targeting almost all SLE-cytokine related receptors, but it is not able to stop signalling of IL-12/IL-23, IL3 and IL-5. Blocking IL12/IL-23

Reviewer 2 Report

This review explore in depth JAK Kinase cytokines network and its role in Systemic Sclerosis (SSc) and in Systemic Lupus Eritematosus (SLE) providing a potential rationale for the use of JAK-Inihibitors.

The use of these drugs , approved for Rheumatoid Arthritis (RA) and Psoriasic Arthritis (PsA),  is one of the hottest topics in Rheumatology. Exploring new potential targets for the treatment of severe diseases such as SSc and SLE it’s extremely important.

The authors describe in a clear and detailed manner the JAK Kinase structure and functions and its potential involvement in SSc and SLE pathogenesis for every single cytokines.

My suggestions for the author are:

1.      Briefly describe JAK-inhibitors family (authors just mention their name describing some studies)

2.      Check language, grammar and punctuation in the whole review. There are some typos in the text. In particular I point out:

·        Row 30  “…is to reduce of…”(typo)

·        Row 50   “…in treatment in many…”(typo)

·        Row 55 “….biologics is sometimes infective….”(please clarify)

·        Row 58   check punctuation

·        Row 108 “…. propagation ….”(please clarify)

·        Row 160 check punctuation

·        Row 205 “…Less in known….” (typo)

·        Rows 231-232 check punctuation and repetition of words

·        Row 307 ” ….corelated…”(typo)

·        Row 316 “….vide…..”(typo)

·        Row 342 “…corelate …”(typo)

·        Row 405 “….a key role seems to be…(please check the verb and the sentence)

·        Row 410 check punctuation

·        Rows 438-439  check the sentence (“but not these…” )and block letters for IL-12 and IL-23

Author Response

Thank you very much for your review

The paper has been  checked again by Dr Laura Coates Professor of Oxford University UK as it was stated in acknowledgements part of the paper

Additionally we corrected some typos and checked punctuation